# AI-Driven De Novo Design and Molecular Modeling for Discovery of Small-Molecule Compounds as Potential Drug Candidates Targeting SARS-CoV-2 Main Protease

**DOI:** 10.3390/ijms24098083

**Published:** 2023-04-29

**Authors:** Alexander M. Andrianov, Mikita A. Shuldau, Konstantin V. Furs, Artsemi M. Yushkevich, Alexander V. Tuzikov

**Affiliations:** 1Institute of Bioorganic Chemistry, National Academy of Sciences of Belarus, 220141 Minsk, Belarus; 2United Institute of Informatics Problems, National Academy of Sciences of Belarus, 220012 Minsk, Belarus; nickshuldov29@gmail.com (M.A.S.); ky6ujlo@gmail.com (K.V.F.); artsemi.yushkevich@gmail.com (A.M.Y.)

**Keywords:** SARS-CoV-2, main protease, deep learning, generative autoencoder, virtual screening, molecular docking, molecular dynamics, binding free energy calculations, anti-SARS-CoV-2 drugs

## Abstract

Over the past three years, significant progress has been made in the development of novel promising drug candidates against COVID-19. However, SARS-CoV-2 mutations resulting in the emergence of new viral strains that can be resistant to the drugs used currently in the clinic necessitate the development of novel potent and broad therapeutic agents targeting different vulnerable spots of the viral proteins. In this study, two deep learning generative models were developed and used in combination with molecular modeling tools for de novo design of small molecule compounds that can inhibit the catalytic activity of SARS-CoV-2 main protease (Mpro), an enzyme critically important for mediating viral replication and transcription. As a result, the seven best scoring compounds that exhibited low values of binding free energy comparable with those calculated for two potent inhibitors of Mpro, via the same computational protocol, were selected as the most probable inhibitors of the enzyme catalytic site. In light of the data obtained, the identified compounds are assumed to present promising scaffolds for the development of new potent and broad-spectrum drugs inhibiting SARS-CoV-2 Mpro, an attractive therapeutic target for anti-COVID-19 agents.

## 1. Introduction

Computer-aided drug design is currently an important tool that can significantly reduce the time and costs required to develop new therapeutic agents [1,2]. In recent years, drug development has increasingly used machine learning methods, in particular deep learning, which are applied at each stage of this complex multi-stage process, not only to accelerate research, but also to assess the risks and costs in clinical trials [3,4]. These methods make it possible to establish mathematical relationships between empirical data on the properties of small molecules and extrapolate them to predict the physicochemical and biological characteristics of new compounds [4,5,6]. Using machine learning, one can explore quantitative structure–activity (QSAR, Quantitative Structure–Activity Relationship) or structure–property (QSPR, Quantitative Structure–Property Relationship) relationships and develop methods that can predict, with high accuracy, the effect of chemical modifications of a compound on its biological activity, pharmacokinetic, and toxicological characteristics [4,5,6]. In addition, machine learning methods can be successfully applied to solve problems of drug repurposing [7,8], protein structure prediction [9], virtual screening of potential drugs, and the prediction of protein−ligand binding affinity [10,11,12]. In recent works, deep learning methods have been used for the screening of chemical databases to identify antibacterial and antiviral inhibitors, including therapeutic agents potentially active against HIV-1 and SARS-CoV-2 [12,13,14,15]. The use of these methods in combination with virtual screening of molecular libraries containing FDA-approved drugs has resulted in the discovery of a number of potential drugs against SARS-CoV-2 [15]. In addition to identifying potential drugs in chemical databases and predicting their physicochemical properties, machine learning is also used to design new compounds with specified pharmacological characteristics [16,17,18,19,20,21]. Despite the fact that the traditional virtual screening of molecular libraries, based on the similarity of the physicochemical characteristics of compounds, provides rich opportunities for identifying novel potential drugs [22], it has certain disadvantages compared to generative models. One of the main incentives for the use of generative models is a broader exploration of the molecular feature space [20]. A similarity-based search provides an exploration of a chemical space limited by the variety of compounds available, while generative models allow for a much broader chemical diversity to be captured in the molecular space of features [20]. The second advantage of generative models is the possibility of imposing additional conditions in the generation process, which enables one to perform the directed design of new molecules from the chemical space instead of their blind generation [14]. The given value of the protein–ligand binding free energy can be used as a criterion, which makes it possible to generate high-affinity molecules from a subset of interest in the studied chemical space [14].

In recent years, deep generative models have found wide application in de novo drug development research [16,17,18,19,20,21]. Thanks to the huge advancements in deep learning methods, generative models with different architectures and different learning methods have now been developed using different types and data structures, including promising models such as the graph neural network [16], recurrent neural network [17], generative adversarial network [18], conditional adversarial autoencoder [19], and generative tensorial reinforcement learning [21]. The application of deep generative models has already shown their ability to generate molecules that can be synthesized, are active in vitro, are stable, and exhibit activity in vivo in models associated with various diseases [15]. In particular, the Janus kinase 3 inhibitor, which presents a new class of immunomodulatory agents, has been developed using the conditionally adversarial autoencoder [19]. In addition, in vivo active inhibitors of discoidin domain receptors 1 and 2 (DDR1 and DDR2) have been developed using generative tensorial reinforcement learning and the pharmacokinetic profile of DDR1 has been confirmed by in vivo mouse experiments [21]. However, despite deep generative models becoming more common in chem- and bioinformatics, their potential in this area has not yet been fully exploited. In this regard, the development and application of deep generative methods for computer-aided drug design are of great scientific and practical importance.

SARS-CoV-2 Mpro plays an important role in mediating viral replication and infectivity and, therefore, is a highly promising therapeutic target [23]. A number of studies have used SARS-CoV-2 Mpro for screening the FDA-approved drugs as potential inhibitors of the virus in the hope of finding drugs effective against COVID-19 [24]. Insights into the literature show that numerous studies are currently underway on natural Mpro inhibitors originating mainly from plants, marine organisms, and microorganisms [24]. Furthermore, covalently-binding peptidomimetics and small molecules are being studied, and some of these various compounds have exhibited antiviral activity in infected human cells [25]. For example, remdesivir, neuraminidase inhibitors, RNA synthesis inhibitors, abidol, and anti-inflammatory drugs have been suggested as potential antiviral agents [26]. A number of potential drug candidates have been found by virtual screening of a library of phytochemicals and Chinese medicinal agents with potential antiviral properties against SARS-CoV-2 Mpro [26]. Studies of Paxlovid (PF-07321332), presenting a nirmatrelvir/ritonavir combination, showed [27] that it blocks SARS-CoV-2 replication by binding to Mpro, is effective when administered orally, and has good selectivity and safety profiles. According to an interim analysis of phase II/III clinical trials, Paxlovid significantly reduced the number of hospitalizations and deaths. In this regard, the clinical trials of Paxlovid were terminated ahead of schedule, and, in November 2021, the U.S. Food and Drug Administration issued an emergency use authorization for the treatment of mild to moderate COVID-19 [28]. Detailed information on the current developments regarding promising inhibitors of SAR-CoV-2 Mpro is presented in the recent review articles [15,24,25,26], testifying that studies on the discovery of new drug candidates against this viral enzyme are still extremely relevant.

The objective of this study consisted of the development of a deep generative neural network, and its application in combination with molecular modeling tools to identify small-molecule compounds able to inhibit the catalytic activity of SARS-CoV-2 main protease (Mpro).

To reach the objective proposed, the following studies were carried out: (i) formation of a training library of small-molecule compounds containing substructures or functional groups that can make the ligand active towards SARS-CoV-2 Mpro; (ii) development and implementation of the architecture of deep generative models based on describing the structures of chemical compounds in the Simplified Molecular-Input Line-Entry System (SMILES) [29] to design potential SARS-CoV-2 Mpro ligands; (iii) training the neural networks followed by validation of the learning outcomes; (iv) generation of a wide range of potential SARS-CoV-2 Mpro ligands using the developed neural networks; and (v) identification of the most promising drug candidates against SARS-CoV-2 Mpro by molecular docking, molecular dynamics simulations, and binding free energy calculations.

## 2. Results and Discussion

An analysis of the data from molecular modeling found the seven best scoring compounds that showed strong attachment to the SARS-CoV-2 Mpro catalytic site in line with the low values of binding free energy predicted both for the static and dynamic ligand/Mpro models. The chemical structures of these top-ranking compounds are shown in Figure 1, and Table 1 and Table 2 shed light on their physicochemical parameters commonly used as the basic filters to screen ligands for their ability to be effective when taken orally. Table 1 indicates that ligand I fully satisfies the requirements imposed on the potential drug by the Lipinski’s “rule of five”, meaning that it possesses highly important characteristics such as absorption, distribution, metabolism, and excretion [30,31]. At the same time, ligands II−VII reveal only one violation of this rule relating to a slight excess of their molecular weight, allowing one to suppose that these compounds also have drug-like properties [30,31]. In addition, the data on a qualitative estimation of the molar solubility of the analyzed compounds (Table 2), which is one of the major properties influencing absorption, suggest that these molecules are soluble in water, evidenced by the values of logS calculated by the ESOL method [32] available on a free web tool SwissADME [33]. Finally, the calculations also indicate that compounds **I**–**VII** (Figure 1) can be synthesized and submitted to biological assays or other experiments, which is a major factor in selecting the most promising virtual molecules. This assumption is supported by the assessment of synthetic accessibility of the molecules of interest using SwissADME (Table 2) which classifies the SA score ranges from 1 (very easy) to 10 (very difficult) [33].

Table 3 and Figure 2 show the profile of interaction modes realized in the docking complexes of the identified compounds with the SARS-CoV-2 Mpro catalytic site. An analysis of the intermolecular interactions of these compounds with Mpro indicates (Table 3, Figure 2) that these ligands form a wide network of van der Waals contacts involving functionally important residues of the Mpro binding pocket, such as His-41, Met-49 (except for compound **VI**), Met-165, Glu-166, and Gln-189. Along with van der Waals interactions, the analyzed compounds constitute hydrogen bonds with Gly-143 (compounds **I**, **III**, **IV**), His-41 (compound **II**), Ser-46 (compound **II**), Thr-24 (compound **VI**), Thr-26 (compounds **II** and **VI**), Glu-166 (compounds **II**, **III**, **IV**, **V** and **VII**), Cys-145 (compound **IV**), His-163 (compound **V**), and Gln-192 (compound **VII**) (Table 3, Figure 2). Furthermore, compounds **III**, **V**, and **VI** participate in specific cation-π interactions with His-41, which is a part of the catalytic dyad of Mpro formed by this residue and Cys-145 [34]. Finally, π-conjugated systems of compounds **VI** and **VII** form π-stacking with the side chain of His-41, and compound **III** makes a salt bridge with Glu-166 (Table 3). Among these binding modes, intermolecular van der Waals interactions and hydrogen bonds are the major contributors to the ligand/Mpro interface (Table 3, Figure 2).

The efficiency of intermolecular interactions in the docking ligand/Mpro complexes is confirmed by the low values of binding free energy and dissociation constant, testifying to the high-affinity binding of compounds **I**–**VII** to the Mpro catalytic site (Table 4). The data of Table 4 suggest that these values calculated using three different scoring functions are at least comparable with those obtained via the identical computational protocol for the control inhibitors **I** and **II**. The MD simulations maintain the major inferences resulting from the analysis of the static ligand/Mpro models. An analysis of the MD trajectories of these models points to their relative stability within the 150 ns time domain, as evidenced by the averages of binding free energies and the corresponding standard deviations (Table 5). Taking into account the MM/GBSA standard error of about 3 kcal/mol [35], the data from molecular dynamics (Table 5) result in the same conclusion made from the results of molecular docking (Table 4), testifying to the similarity in the binding affinity profiles of the predicted and control compounds. This conclusion is also indicated by the data on the time dependences of the root-mean-square deviations (RMSD) of the atomic positions between all of the MD complexes and their starting models (Figure 3). For the predicted compounds, the averages of RMSD, equal to 2.14 ± 0.31 Å (compound **I**), 2.39 ± 0.24 Å (compound **II**), 2.05 ± 0.32 Å (compound **III**), 2.26 ± 0.28 Å (compound **IV**), 1.98 ± 0.23 Å (compound **V**), 2.39 ± 0.33 Å (compound **VI**), 1.80 ± 0.24 Å (compound **VII**), are close to the mean values of 1.94 ± 0.30 Å and 1.99 ± 0.27 Å calculated for the control inhibitors **I** and **II**, respectively (Figure 3). At the same time, the mean value of the RMSD for SARS-CoV-2 Mpro in the unbound state is 1.89 ± 0.30 Å, indicating that this average is comparable with those calculated for the predicted compounds bound to the enzyme (Figure 3). This is an additional confirmation that the ligand/Mpro complexes do not undergo significant structural reorganizations on their MD trajectories. Finally, the relative stability of the analyzed complexes is encouraged by the data on the time dependences of the values of binding free energy, which show no tendency to increase over time (Figure 4).

Insights into the data on the contributions of the individual Mpro amino acids to the binding enthalpy reveal the residues dominating the ligand/Mpro interface. Table 6 indicates that these residues are His-41 (compounds **I**–**V**, **VII**), Met-49 (compounds **I**–**VII**), Asn-142 (compounds **III**–**VI**), Gly-143 (compounds **III**–**VI**), Cys-145 (compounds **III**–**VII**), Met-165 (compounds **I**–**V**, **VII**), Glu-166 (compounds **I**, **III**–**VII**), Asp-187 (compounds **I**, **II**, **VII**), and Gln-189 (compounds **I**–**V**, **VII**) (Table 6). For the control inhibitors **I** and **II**, the common binding hot spots to the SARS-CoV-2 Mpro are Met-49, Asn-142, Gly-143, Cys-145, Met-165, Glu-166, Asp-187, and Gln-189 (Table 6). In addition, inhibitor **I** also participates in strong binding to the Mpro His-41 residue. Importantly, most of these residues are used by the predicted compounds for effective interaction with the SARS-CoV-2 Mpro. The data obtained suggest that these major contributors to the ligand/Mpro interaction play the role of anchor residues, providing strong attachment of the identified and control compounds to the enzyme catalytic site. Among these binding hot spots, the highly important His-41 and Cys-145 should first be noted, forming the catalytic dyad of SARS-CoV-2 Mpro [34].

The calculation of the root-mean-square fluctuations (RMSF) of the individual Mpro residues, testifying to the flexibility of each amino acid on the MD trajectory, indicates that most of the enzyme residues show small structural fluctuations (Figure 5). Moreover, this observation applies both to the residues of Mpro in the complexes with the predicted compounds and control inhibitors, and to those of the enzyme in the unbound state (Figure 5). The averages of RMSF values for the ligand/Mpro models and unliganded enzyme are about the same, ranging between 0.88 Å and 1.15 Å. Importantly, the Mpro residues dominating the ligand/Mpro interface are also positionally restrained (Table 7). For these anchor residues, the RMSF values do not exceed 2.2 Å, and, in most cases, are less than 1.5 Å (Table 7). At the same time, the RMSF values calculated for these hot spots of the enzyme bound to the identified and control compounds are close to each other (Table 7), supporting their key role in the ligand/Mpro interaction.

Thus, the data on the binding affinity profiles of the identified compounds obtained using molecular docking and molecular dynamics tools are in agreement with each other, indicating the strong attachment of these ligands to the Mpro catalytic site (Table 4 and Table 5). According to these data, the high-affinity binding of the predicted molecules to the catalytic pocket of Mpro is generally provided by hydrogen bonds and van der Waals dispersion forces (Table 2, Figure 3), which are implemented by the relevant pharmacophore groups present in their structures, namely by donors and acceptors of H-bonds and non-polar chemical motifs (Figure 1, Table 1). The computational findings indicate that it is these pharmacophore groups that make a crucial contribution in providing the analyzed molecules with the ability to inhibit the catalytic activity of Mpro. This suggests that the used computational approach combining AI-driven de novo design with molecular modeling allowed one to avoid false-positive results and properly evaluate the strength of intermolecular interactions. This supposition is indirectly confirmed by the findings of a recent study [37] in which the use of the machine-learning scoring function NNScore 2.0 in combination with one to four classical scoring functions was shown to provide the best accuracy of binding affinity prediction. Taken together, the data obtained give strong evidence to assume that the predicted compounds may exhibit low values of binding free energy to Mpro, close to those calculated for the control inhibitors **I** and **II**. Based on these data, it can be expected that small-molecule compounds **I**−**VII** (Figure 1) have good therapeutic potential to inhibit the enzyme catalytic activity, and, therefore, may serve as good scaffolds for the drug developments targeting SARS-CoV-2 main protease. However, it is important to note that these molecules can have a number of undesirable physicochemical characteristics, which may limit their therapeutic application. In particular, these characteristics include important pharmacological properties for drugs such as cytotoxicity, solubility in aqueous systems, and bioavailability, which cannot always be predicted correctly by computational methods. Obviously, only biomedical research of the identified compounds can validate the data from molecular modeling, and therefore the further advancement of this work assumes the implementation of the following stages: (i) synthesis and in vitro testing of the predicted molecules for antiviral activity, cytotoxicity, and mechanism of action; (ii) identification of the lead compounds and their optimization using the current QSAR strategies [38]; (iii) synthesis of the optimized compounds followed by detailed biomedical assays.

## 3. Materials and Methods

### 3.1. Development of a Deep Generative Neural Network

#### 3.1.1. Preparing the Training Dataset

The generative autoencoder was constructed to be specific for SARS-CoV-2 Mpro, and, therefore, the training dataset for the neural network should have included compounds potentially active against this therapeutic target. For this reason, a virtual molecular library of potential anti-SARS-CoV-2 agents able to effectively bind and block the Mpro catalytic site was primarily formed for learning of the neural network. The procedure for preparing this library is described below.

(A)Building pharmacophore models and virtual screening of chemical databases

To identify small-molecule compounds potentially active against SARS-CoV-2 Mpro, the pharmacophore-based virtual screening was performed by a web-oriented platform Pharmit (http://pharmit.csb.pitt.edu (accessed on 25 April 2023)) [39], allowing one to search for small molecules based on their structural and chemical similarity to another small molecule. To do this, pharmacophore models that described a set of structural and functional features providing high-affinity binding to Mpro for a number of potent inhibitors of SARS-CoV, the predecessor of SARS-CoV-2 [23], were constructed. Chemical structures of these SARS-CoV inhibitors were taken from a study [23] and divided into groups according to their belonging to a certain class of chemical compounds, and then the pharmacophore characteristics of representative members of each group were averaged using the PharmaGist web server [40]. As a result, 16 pharmacophore models that corresponded to 16 groups of the SARS-CoV Mpro inhibitors including 6 classes of peptidomimetics and 10 classes of small-molecule compounds [23] were built. The obtained dataset was then supplemented with the pharmacophore model of the pan-coronavirus inhibitor X77, a potent non-covalent antiviral agent targeting Mpro both of SARS-CoV, MERS-CoV, and SARS-CoV-2 [41,42]. The X77 pharmacophore model was generated using the web-platform Pharmit (http://pharmit.csb.pitt.edu) [39] based on the structure of the X77/Mpro complex in crystal (PDB ID: 6w63; https://www.rcsb.org (accessed on 25 April 2023)) [42]. Pharmacophore-based virtual screening was performed in the nine Pharmit molecular libraries containing over 213.5 million chemical structures, resulting in a set of 711,102 compounds that satisfied one of the seventeen constructed pharmacophore models.

(B)Molecular docking

Compounds identified by the pharmacophore-based screening were subject to the preliminary molecular docking with the SARS-CoV-2 Mpro structure (PDB ID: 6Y84; https://www.rcsb.org/pdb/ (accessed on 25 April 2023)) [42]. These compounds were then filtered via the docking scoring function with the energy threshold value of −7 kcal/mol, which corresponds to the IC_50_ value of 10 μM generally used in the in vitro screening of potential drugs [43]. At the next stage, the refining molecular docking of the SARS-CoV-2 Mpro structure with 353,467 compounds that met the given threshold value of binding free energy was carried out. As a result, 347,732 compounds with values of binding energy lower than −6 kcal/mol were selected for further analysis. The higher energy threshold for the refining docking compared to the preliminary docking was chosen based on two major factors. First, the AutoDock Vina program which was used for the refining docking has a higher accuracy in predicting the ligand-binding affinity than QuickVina 2 (Oxford, UK) [44], applied for the preliminary docking. Second, for the refining docking, a much larger number of ligand poses in the Mpro catalytic site were considered (see Section 3.1.6. below). The significant reduction in the size of the training dataset, achieved using two-stage docking, made it possible to essentially decrease the number of parameters of the developed neural network and thereby speed up the process of its training.

#### 3.1.2. SMILES Space Revision and Vectorization

The generated library of chemical compounds in the SMILES linear notation was purified from molecules that had a molecular weight more than 1000 Da, duplicates, and incorrect SMILES records. The representations of chemical structures in the SMILES format were obtained by the Python 3 script using the RDKit module [45]. Molecules containing at least one SMILES element with a frequency less than 0.001 were filtered out based on the frequency distribution of SMILES elements in the generated dataset. The distribution of SMILES lengths was then analyzed, and compounds with the SMILES representation longer than 120 characters were removed (Figure 6A). After applying all the filters, the dataset included 342,102 different molecules and corresponding SMILES. The SMILES were vectorized into a matrix in accordance with the maximum length and symbols’ vocabulary size (Figure 6B), with the added start and end symbols represented by “!” and “E”.

The resulting 342,102 molecules integrated with the corresponding values of docking scores constituted the dataset, which was divided into the training, validation, and test subsets involving 70%, 15%, and 15% of the initial dataset, respectively. While generating the subsets, a stratified division was applied to retain the same energy distributions within all three sets. The validation set was used to assess the model’s power to reconstruct the input SMILES during training, whereas the test SMILES were employed to sample novel molecules by the addition of distortion to their latent representation.

#### 3.1.3. Restoration of Three-Dimensional Structures of Generated Molecules

To estimate the capability of the neural network to generate new compounds active against SARS-CoV-2 Mpro, a molecular docking of these molecules with the enzyme needed to be performed. Obviously, this required the use of three-dimensional (3D) structures of the generated molecules. To obtain these 3D structures from the SMILES format, a script was designed in Python 3 using the RDKit module [45]. The process of structure generation comprised the following stages: SMILES input, SMILES validity check, generation of 2D atomic coordinates, generation of 3D atomic coordinates, structure optimization in the MMFF94 force field [46,47], attachment of hydrogen atoms, and re-optimization in the MMFF94 force field. The generation of 3D atomic coordinates of molecules was carried out by the ETKDGv3 algorithm [48].

#### 3.1.4. Preparation of the Mpro Structure

The X-ray structure of the SARS-CoV-2 Mpro in the unbound state was accepted from the Protein Data Bank (PDB ID: 6Y84; https://www.rcsb.org/pdb/) [42]. Hydrogen atoms were then added to this structure, followed by annotating atoms with Gasteiger partial charges [49], and structure optimization in the UFF force field [50]. To achieve this, the OpenBabel program [51] was used. The obtained structure of SARS-CoV-2 Mpro was applied for the preliminary and refining docking, both during the dataset preparation and the docking of the generated compounds.

#### 3.1.5. Preparation of Ligand Structures

The preparation of the ligand structures for the preliminary docking was the same as reported for the SARS-CoV-2 Mpro structure. This procedure was carried out by OpenBabel, but included an additional step of rotatable bond identification, which is auto-made by this program [51]. At the same time, the preparation of the ligand structures for the refining docking was performed using the following two steps: (i) optimizing in the MMFF94 force field [46,47] for removing steric clashes and the addition of hydrogen atoms missing in the original structures by the RDKit module [45] in Python 3; (ii) addition of Gasteiger partial charges and rotatable bonds identification using MGLTools [52]. It is important to note that before molecular docking, the 3D structures of new ligands were obtained from the generated linear SMILES notations, as described above.

#### 3.1.6. Computational Protocol of Molecular Docking

The preliminary and refining molecular docking were carried out by QuickVina 2 [44] and AutoDock Vina (La Jolla, CA, USA) [53], respectively, in the approximation of rigid receptor and flexible ligands. In both cases, the grid box contained the catalytic pocket of SARS-CoV-2 Mpro and had the following parameters: ΔX = 19 Å, ΔY = 21 Å, ΔZ = 23 Å centered at X = −20 Å, Y = 19 Å, Z = −26 Å. The value of the exhaustiveness parameter setting the number of individual sample “runs” was equal to 10 and 50 for preliminary and refining docking, respectively.

#### 3.1.7. Architectures of Deep Generative Models

Two deep generative models were designed, an unsupervised SMILES-based Long Short-Term Memory (LSTM) autoencoder [54] (the embeddings model) and a semi-supervised SMILES-based LSTM autoencoder (the energy model) [55]. The value of binding free energy was employed in the energy model as an additional parameter in the latent layer for learning compounds from the training set based on the results of molecular docking, and as the desired value of their binding affinity to Mpro in the mode of generating new compounds. The high-level architectures representative of these two models were integrated into one scheme, as shown in Figure 7.

The embeddings model includes the encoder part, 2D Gaussian noise entry point on the latent layer, and the decoder part. This model receives the vectorized SMILES matrix as an input that passes through the LSTM layer. The feature of this model consists in the fact that the LSTM output itself is not used. Instead, the hidden and cell states’ vectors are derived and concatenated together, and then they move through a dense layer. The output of this dense layer presents a latent vector or SMILES embeddings which are submitted to two dense layers in parallel, making initial hidden and cell state inputs for the LSTM layer in the decoder part. The decoder input layer in the training regime receives the same vectorized SMILES matrix *X* as the encoder input, and, as a traditional LSTM generative model, it predicts the next symbol. The decoder input launches the process of generation with a start symbol ‘!’ only; embeddings are used to predict the initial states of decoder LSTM, and they mainly determine which kind of SMILES will be obtained.

Unlike the embeddings model, the energy model has the additional neuron on the latent layer responsible for the value of binding free energy. Whereas the embeddings model can generate compounds from random SMILES embeddings and add noise to SMILES embeddings of ligands with predicted binding free energy, the energy model can generate novel ligands with a preset binding free energy, in addition to efforts to manipulate the SMILES embeddings of the ligands to try to improve their structures after decoding and thus decrease binding free energy.

#### 3.1.8. Training the Models

Both models were constituted layer by layer by TensorFlow 2.1 [56] and exposed to 150 epochs of training; additionally, “Reduce learning rate on a plateau” and “Early stopping” callbacks were applied to assist the model’s convergence to a better local minimum and also to avoid overfitting. The optimizer used a method for stochastic optimization Adam [57], with the 0.005 learning rate initial value. At the same time, the categorical cross-entropy loss function [58] was used. The loss functions for both models are shown in Figure 8.

#### 3.1.9. Deep Learning-Based Compounds’ Generation

Two modes of generation have been investigated in our recent report [59]. The first mode was the generation from random numbers taken from normal distributions, where distribution parameters were obtained using test data distribution on the latent layer for each vector component (‘pass’ mode in the operator ‘add/pass’, Figure 7). In this case, the generation process for the energy model involved setting an a priori value of binding free energy to approximate the generated compounds. The main feature of the second mode of generation consisted of sampling the best ligands from the test set, and trying to add noise to their SMILES embeddings (‘add’ mode in the operator ‘add/pass’, Figure 7). This approach was assumed to change the reconstructed ligand, and, for the energy model, also improve the binding free energy, making the neural network generate more prospective compounds. The combinations of two autoencoder models and two generation modes are given in Table 8.

The data on the validation of the autoencoder model’s work in two generative modes have been previously published in a study [59]. According to this study, the developed neural network has a great potential to enrich screening pipelines with new small-molecule compounds able to inhibit the catalytic site of Mpro. Importantly, the embeddings and energy models using only SMILES representation of the reference compounds randomly selected from the test set in various generation modes showed the best efficiency of the neural network operation [59]. It is for this reason that these combinations of models and generation modes were used in the present work for de novo design of potential drug candidates against COVID-19.

### 3.2. De Novo Design of Potential Inhibitors Targeting SARS-CoV-2 Mpro

#### 3.2.1. Generation of a Wide Set of Potential SARS-CoV-2 Mpro Ligands

Using the developed neural network, a wide set of potential ligands of SARS-CoV-2 Mpro was generated for the subsequent identification of promising inhibitors of this critically important viral enzyme by molecular modeling methods. As noted above, both embeddings and energy models of the autoencoder were used to generate the ligands, and, in both cases, only embeddings of the SMILES representations of the reference compounds were used as input data. As a result of the autoencoder operation, the SMILES linear representations for 128,955 molecules were obtained. A reconstruction of the 3D molecular structures from their SMILES linear representations was performed using an algorithm that included the following steps: (1) reading the structure and generating 2D atomic coordinates for molecules from the SMILES descriptions; (2) generation of 3D atomic coordinates of molecules; (3) addition of hydrogen atoms; (4) optimization of the generated 3D molecular structures; (5) verification of the maintenance of the molecule’s initial stereochemistry in the final structures. The total number of molecules with the correct 3D structures that were selected for building their complexes with SARS-CoV-2 Mpro by molecular docking was 95,775.

#### 3.2.2. Molecular Docking of the Generated Compounds with SARS-CoV-2 Mpro

The generated compounds were prepared for molecular docking using the MGLTools software (La Jolla, CA, USA) [52]. Molecular docking of these compounds with the crystal SARS-CoV-2 Mpro structure (PDB ID: 6Y84) [42] was carried out by the AutoDock Vina program [53] in the approximation of a rigid receptor and flexible ligands. The grid box parameters were the same as those described above in the subsection “Computational protocol of molecular docking”. The value of the exhaustiveness parameter was set to 100. The values of binding free energy were then estimated in terms of three scoring functions, namely AutoDock Vina (https://vina.scripps.edu (accessed on 25 April 2023)) [53], RF-Score-4 (https://pjballester.wordpress.com/software/ (accessed on 25 April 2023)) [37], and NNScore 2.0 (https://git.durrantlab.pitt.edu/jdurrant/nnscore2 (accessed on 25 April 2023)) [60]. According to the values of each scoring function, the ranks of compounds were calculated and the value of the exponential consensus ranking (ECR) function was then obtained from these data for each ligand by the following formula [61]:$$ECR=\sum_{sf}\frac{1}{\sigma_{sf}}*exp\{-\frac{{rank}_{sf}}{\sigma_{sf}}\}$$
where ${rank}_{sf}$ is the rank of the compound according to the scoring function *sf*, $\sigma_{sf}$ is the parameter that controls the influence of the scoring function *sf* on the results of consensus selection (*ECR* was calculated using $\sigma_{sf}$ = 10 for all considered *sf*, since the contributions of the individual scoring functions were taken equal).

The analyzed compounds were ranked based on the consensus *ECR* ranking, and ligands with the same rank were assigned to separate groups. As a result, the ligands that belonged to 11 groups with the highest *ECR* values were selected out of 95,775 initial compounds, allowing 39 compounds to be identified. The complexes of these 39 compounds with SARS-CoV-2 Mpro were analyzed using molecular dynamics (MD) simulations and binding free energy calculations.

#### 3.2.3. Molecular Dynamics Simulations

The preparation of the ligand/Mpro complexes for molecular dynamics simulations was performed by the AmberTools18 software (San Francisco, CA, USA) [62]. Molecular dynamics was carried out using Amber18 in the Amber ff14SB (Mpro) and GAFF (ligands) force fields [62]. The Antechamber module was used to calculate the AM1-BCC atomic partial charges [62]. The general Amber force field [63] was employed for the preparation of the force field parameters. Hydrogen atoms were attached to Mpro via the tleap program of AmberTools18 [62]. The ligand/Mpro complexes were each put in a cubical box with periodic boundary conditions. The box for the MD simulations also contained the TIP3P water model [64] as an explicit solvent with Na^+^ and Cl^−^ ions ensuring an overall salt concentration of 0.15 M. An energy minimization of the complex assembly was then made by 500 steps of the steepest descent algorithm and 500 steps of the conjugate gradient method. After energy minimization, the backbone atoms of the system were constrained by an additional harmonic potential, with a force constant of 2.0 kcal/mol, and the complex assembly was exposed to the equilibration phase. The system equilibration was implemented in three sequential stages including (i) the system heating from 0 K to 300 K for 1 ns in NVT ensemble by a Langevin thermostat with a collision frequency of 2.0 ps^−1^ [62]; (ii) pressure equilibration during 1 ns at 1.0 bar in NPT ensemble using a Berendsen barostat with a 2.0 ps characteristic time [62]; (iii) removal of the constraints imposed on the complex assembly and its re-equilibration at 300 K over 0.5 ns under constant volume. The MD simulations were then made during 200 ns in NPT ensemble at temperature T = 300 K and P = 1 bar. Bonds with hydrogen atoms were constrained by the SHAKE algorithm [65] to reach the integration time-step of 2 ps. Long-range electrostatic interactions were calculated using the Particle Mesh Ewald (PME) algorithm [66]. Coulomb interactions and van der Waals interactions were truncated at 8 Å.

#### 3.2.4. Analysis of Interaction Modes and Binding Affinity Profile

The interaction modes of the predicted compounds to Mpro appearing in the docked/ligand complexes were identified using the BINANA software [67]. The ligand poses in the Mpro catalytic site were visualized by the molecular visualization system PyMOL [68].

The values of binding free energy were predicted using Amber18 (San Francisco, CA, USA) [62] by the MM/GBSA method [35,69,70]. The calculations were carried out for 150 snapshots derived from the final 150 ns of the MD trajectories, by keeping the snapshots every 1.0 ns. The polar solvation energies were calculated in continuum solvent using the Poisson–Boltzmann continuum solvation model with an ionic strength of 0.15. The non-polar terms were evaluated via solvent accessible surface area [62]. The values of the entropic term were obtained using the Nmode module of AmberTools18 [62]. An analysis of the MD trajectories was performed for the last 150 ns of the MD simulations using the CPPTRAJ module of AmberTools18 [62].

Two potent SARS-CoV-2 Mpro inhibitors with the IC_50_ values of 18 ± 2 nM and 20 ± 5 nM (compounds 21 and 23 in a study [71], respectively) were used in the calculations as a positive control. These molecules are among the strongest non-covalent inhibitors of Mpro currently known [71] and are therefore very suitable for use as reference compounds in the selection of promising drug candidates targeting the catalytic site of this SARS-CoV-2 enzyme. These control compounds are denoted below as inhibitors **I** (IC_50_ = 18 ± 2 nM) and **II** (IC_50_ =20 ± 5 nM).

## 4. Conclusions

In the present study, two deep generative models were developed and applied for the computer-aided development of novel potential inhibitors targeting SARS-CoV-2 main protease, an enzyme critically important for mediating viral replication and transcription [23]. The training and testing of these models for two generation modes were carried out and the results of their operation were evaluated. The developed neural network was shown to have good potential to generate new compounds with the preset antiviral potency. The generative models were used for de novo design of a wide set of potential ligands of SARS-CoV-2 Mpro, which resulted in 95,775 unique chemical structures. These structures were then screened by molecular docking and molecular dynamics tools to identify the most probable inhibitors of the enzyme catalytic activity. The calculations revealed the seven top-ranking compounds that showed high-affinity binding to Mpro, in agreement with the low values of binding free energy, RMSD, and RMSF. Importantly, the averages of binding free energy predicted for these compounds turned out to be comparable with those calculated for two potent inhibitors of Mpro [71] using the same computational protocol. In light of the data obtained, the identified compounds are assumed to present promising basic structures for the development of new potent and broad-spectrum drugs inhibiting an attractive therapeutic target for anti-COVID-19 agents.

## Figures and Tables

**Figure 1 ijms-24-08083-f001:**
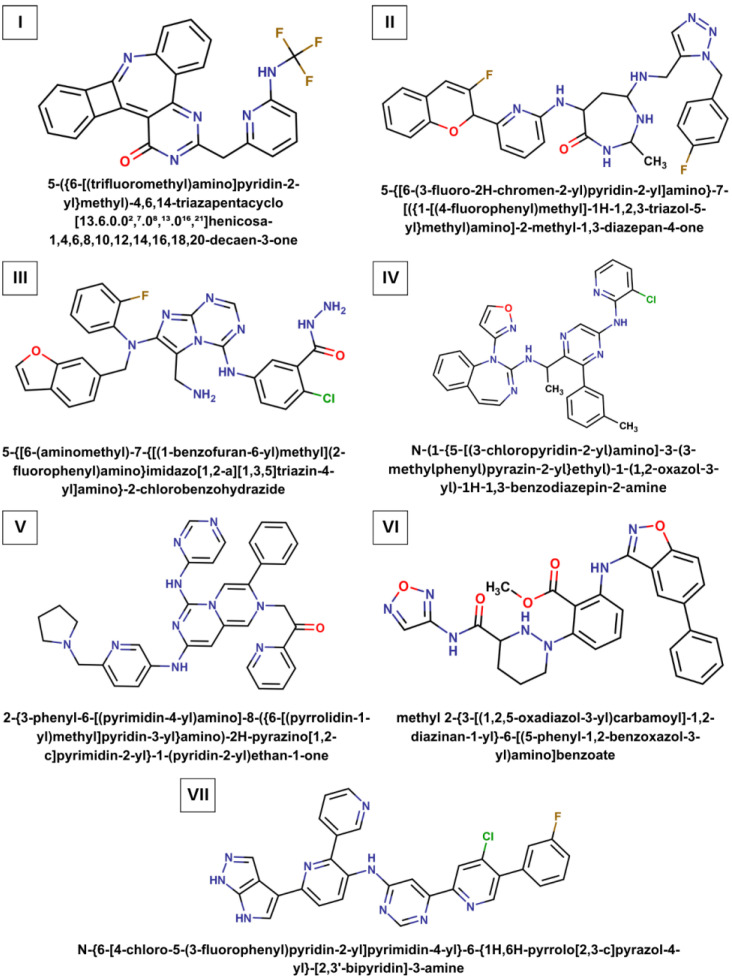
Chemical structures of the identified compounds. The systematic names of these molecules are given.

**Figure 2 ijms-24-08083-f002:**
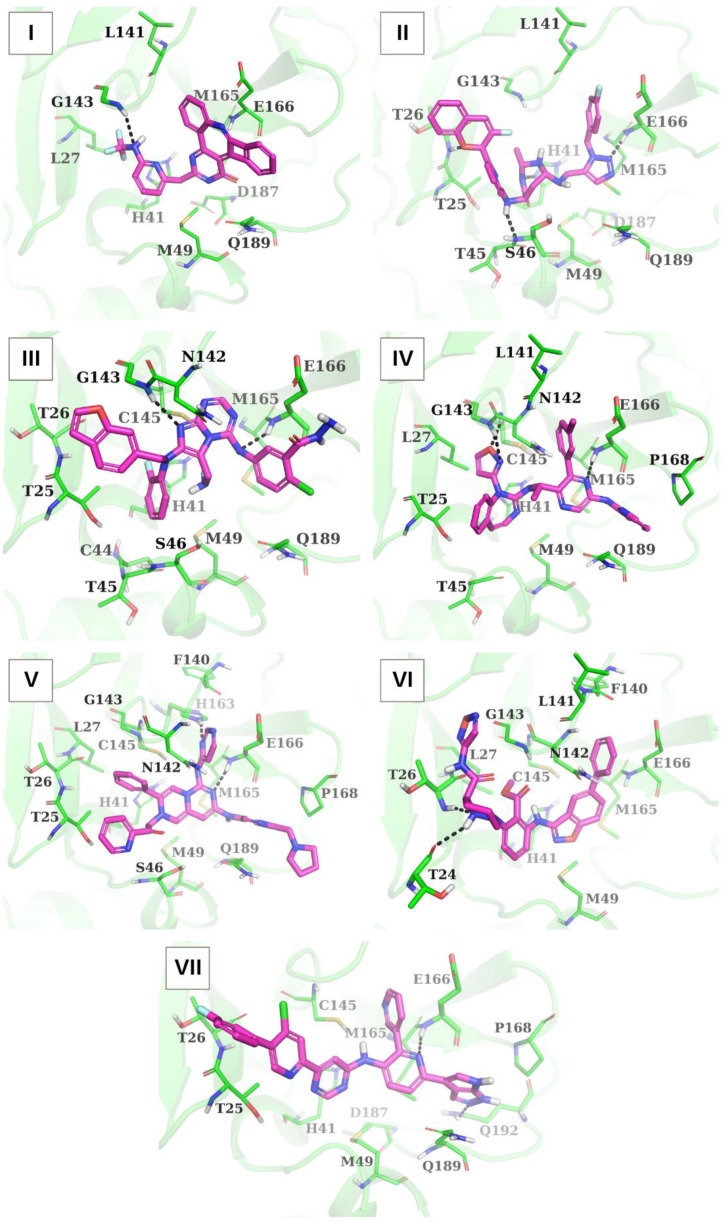
Structural complexes of compounds **I**–**VII** with SARS-CoV-2 Mpro generated by molecular docking. The enzyme residues forming intermolecular contacts with the ligands are indicated. Hydrogen bonds are shown by dashed lines.

**Figure 3 ijms-24-08083-f003:**
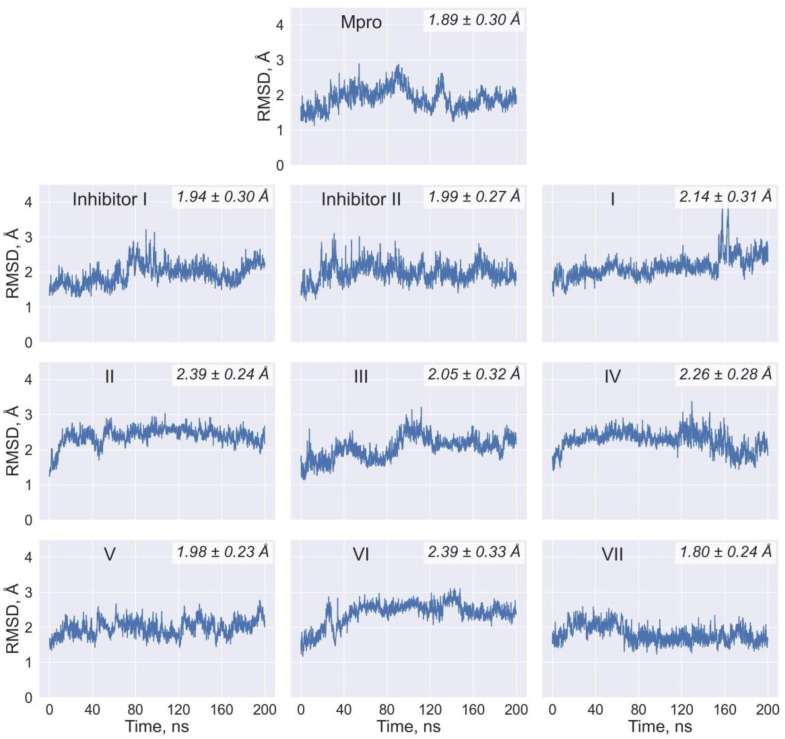
The time dependence of the RMSD (Å) calculated between all of the MD structures and the starting models of the identified and control compounds bound to SARS-CoV-2 Mpro. The corresponding data are also shown for the enzyme in the unbound state. In the upper right corner, the mean values of RMSD and corresponding standard deviations for the last 150 ns of the MD trajectories are indicated.

**Figure 4 ijms-24-08083-f004:**
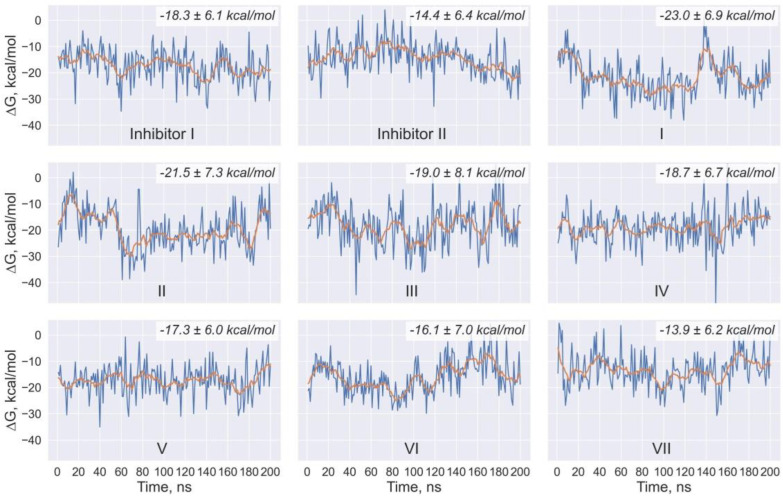
Time dependences of binding free energies for complexes of the identified and control compounds with SARS-CoV-2 Mpro. The orange line shows a simple moving average with a window size of 20 ns. In the upper right corner, the mean values of binding free energies and corresponding standard deviations calculated for the last 150 ns of the MD trajectories are indicated.

**Figure 5 ijms-24-08083-f005:**
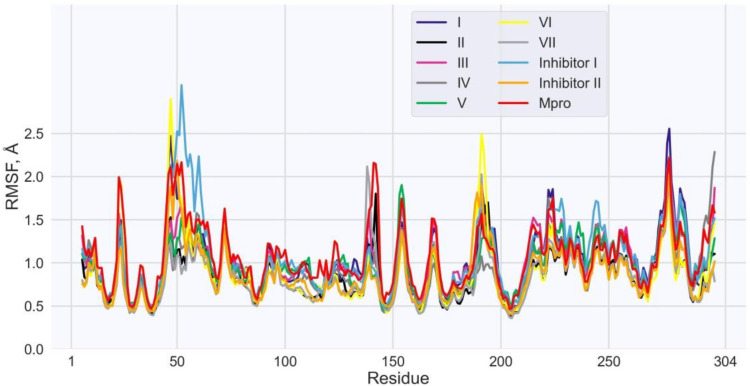
Values of RMSF (Å) for each residue along the Mpro amino acid sequence. The following designations are used: purple, black, pink, grey, green, yellow, and light grey lines correspond to Mpro in the complexes with compounds **I**, **II**, **III**, **IV**, **V**, **VI**, **VII**, respectively; blue and orange lines match Mpro bound to the control inhibitors **I** and **II**, respectively; red line satisfies Mpro in the unbound state.

**Figure 6 ijms-24-08083-f006:**
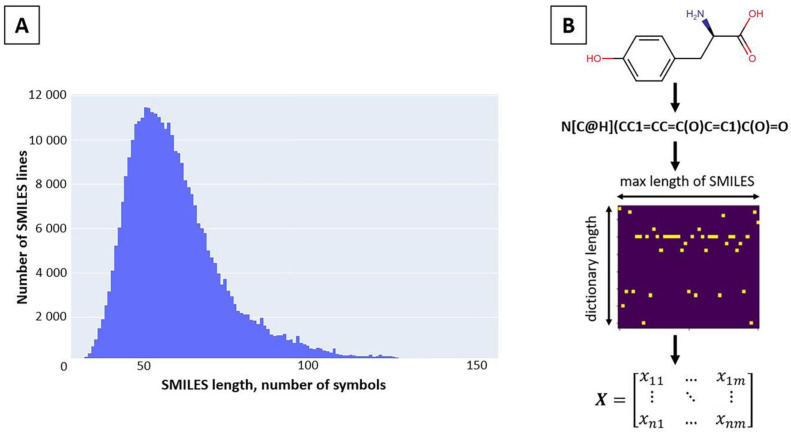
SMILES space revision and vectorization: (**A**) SMILES length distribution for the training set; (**B**) vectorization procedure of a molecule.

**Figure 7 ijms-24-08083-f007:**
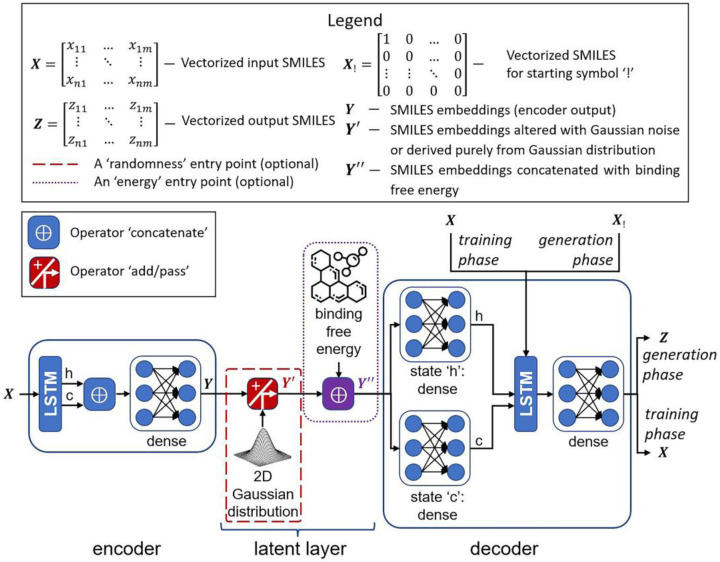
High-level architecture of molecular autoencoder models.

**Figure 8 ijms-24-08083-f008:**
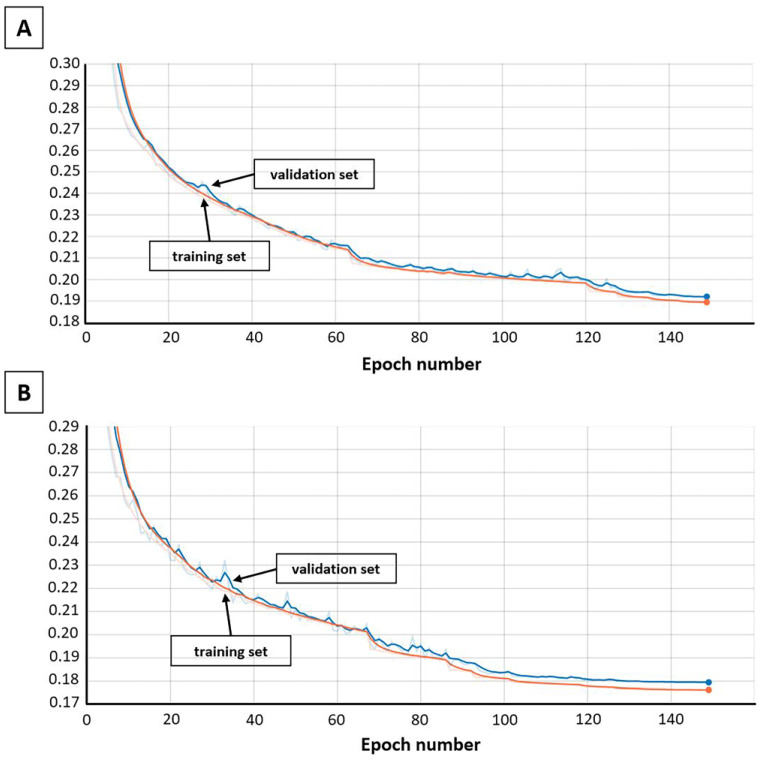
Train and validation losses for autoencoder model: (**A**) unsupervised (embeddings) model; (**B**) semi-supervised (energy) model.

**Table 1 ijms-24-08083-t001:** Physicochemical parameters of the identified compounds associated with the Lipinski’s “rule of five”.

Ligand	Chemical Formula	Molecular Weight (Da)	LogP	Number of H-Bond Donors	Number of H-Bond Acceptors
**I**	C_25_H_14_F_3_N_5_O	457.4	4.43	1	8
**II**	C_30_H_30_F_2_N_8_O_2_	572.6	2.93	4	9
**III**	C_28_H_23_C_l_FN_9_O_2_	572.0	3.48	5	8
**IV**	C_30_H_25_C_l_N_8_O	549.0	4.73	2	6
**V**	C_34_H_32_N_10_O	596.7	3.62	2	7
**VI**	C_28_H_25_N_7_O_5_	539.5	3.43	3	9
**VII**	C_30_H_19_C_l_FN_9_	560.0	4.93	3	7

Physicochemical parameters were calculated using a freely accessible web-server SwissADME [33].

**Table 2 ijms-24-08083-t002:** Data on the molar solubility in water and synthetic accessibility predicted for the designed compounds by the SwissADME web server.

Ligand	Decimal Logarithm of the Molar Solubility in Water LogS	Synthetic Accessibility SA
**I**	−5.68	3.55
**II**	−5.40	5.70
**III**	−6.37	4.07
**IV**	−6.62	5.23
**V**	−6.79	4,52
**VI**	−6.28	4.83
**VII**	−6.75	3.89

**Table 3 ijms-24-08083-t003:** Intermolecular interactions appearing in the docking complexes of the identified compounds with SARS-CoV-2 Mpro.

Ligand	Hydrogen Bonds ^1^	Van Der Waals Contacts ^2^	Cation-π Interactions, Salt Bridges, and π-π Stacking ^3^
**I**	N...*HN[G143]	E166(5), M49(3), L141(2), H41(2), M165(1), Q189(1), L27(1)	−
**II**	NH...**N[H41] NH...*N[S46] O...*HN[T26] N...* HN[E166]	E166(7), M165(3), T25(2), T26(2), L141(2), T45(1), H41(1), M49(1), S46(1), G143(1)	−
**III**	N...*HN[G143] N...*HN[E166]	E166(4), N142(3), T25(3), T45(2), M49(2), T26(1), S46(1), C44(1), H41(1), M165(1), Q189(1)	H41(2) (cation-π interaction); E166 (salt bridge)
**IV**	N...*HN[E166] O...*HN[C145] N...*HN[G143]	E166(8), L27(6), Q189(6), T25(3), L141(3), P168(3), H41(1), M49(1), T45(1), G143(1), C145(1), M165(1), N142(1)	−
**V**	N...*HN[E166] N...**HN[H163]	Q189(7), P168(6), T25(5), E166(4), F140(2), S46(2), L27(2), G143(2), M49(1), M165(1), H41(1), T26(1)	H41 (cation-π interaction)
**VI**	NH...*O[T24] N...*HN[T26]	E166(8), F140(3), L141(3), G143(3), M165(2), H41(1), L27(1), C145(1)	H41 (cation-π interaction); H41 (π-π stacking)
**VII**	N...*HN[E166] N...**HN[Q192]	Q189(8), E166(5), T25(4), M165(3), H41(2), P168(2), M49(1), T26(1)	H41 (π-π stacking)

^1^ Atoms of the ligands are shown first, followed by the corresponding atoms of SARS-CoV-2 Mpro (Mpro residues are in brackets in one-letter code). Symbol * denotes the atoms of the residue main chain, and symbol ** marks the atoms of the residue side chain. ^2^ Amino acids of SARS-CoV-2 Mpro forming van der Waals contacts with the ligands. The number of contacts is given in brackets. ^3^ For cation-π interactions, π-π stacking, and salt bridges, residues of SARS-CoV-2 Mpro involved in these binding modes are denoted.

**Table 4 ijms-24-08083-t004:** Values of binding free energy (Δ*G*) and dissociation constant (K_d_) for the static ligand/Mpro complexes according to the scoring functions AutoDock Vina, RFScore4, and NNScore2.

Ligand	ΔG_VINA_ ^1^ kcal/mol	K_dVINA_ ^1^ μM	ΔG_RFScore4_ ^2^ kcal/mol	K_dRFScore4_ ^2^ μM	ΔG_NNScore2.0_ ^2^ kcal/mol	K_dNNScore2.0_ ^2^ μM
**I**	−9.1	0.384	−10.9	0.022	−11.9	0.0041
**II**	−10.3	0.055	−11.0	0.016	−11.6	0.0069
**III**	−8.7	0.735	−11.1	0.015	−12.7	0.0012
**IV**	−10.0	0.089	−11.1	0.014	−13.0	0.0007
**V**	−9.2	0.326	−10.9	0.022	−13.4	0.0004
**VI**	−9.6	0.171	−11.2	0.012	−11.9	0.0043
**VII**	−9.9	0.105	−11.2	0.012	−12.9	0.0007
Inhibitor **I**	−8.3	1.407	−11.0	0.018	−8.1	1.9
Inhibitor **II**	−8.5	1.017	−11.1	0.015	−7.9	2.9

^1^ The values of ΔG predicted by AutoDock Vina were converted to those of K_d_ using the formula ∆G = R × T × ln(K_d_) (where ∆G is the binding free energy, R is the universal gas constant, T is the absolute temperature equal to 310 K) [36]. ^2^ This formula was also used to convert the values of K_d_ estimated by RFScore 4 and NNScore 2.0 to those of ∆G.

**Table 5 ijms-24-08083-t005:** Averages of binding free energy (<ΔG>) for the dynamic complexes of the identified ligands and control compounds with SARS-CoV-2 Mpro and their standard deviations (ΔG_STD_) calculated for the final 150 ns of the MD trajectories.

Ligand	<ΔH> kcal/mol	ΔH_STD_ kcal/mol	<TΔS> kcal/mol	(TΔS)_STD_ kcal/mol	<ΔG> kcal/mol	ΔG_STD_ kcal/mol
**I**	−45.3	5.2	−22.3	4.5	−23.0	6.9
**II**	−46.0	6.2	−24.4	4.7	−21.5	7.3
**III**	−46.6	6.0	−27.6	6.0	−19.0	8.1
**IV**	−44.6	3.9	−25.9	5.3	−18.7	6.7
**V**	−45.5	4.3	−28.2	4.3	−17.3	6.0
**VI**	−40.9	6.2	−24.8	4.4	−16.1	7.0
**VII**	−37.4	4.8	−23.4	3.9	−13.9	6.2
Inhibitor **I**	−42.8	4.1	−24.5	4.9	−18.3	6.1
Inhibitor **II**	−39.0	4.1	−24.5	4.5	−14.4	6.4

<ΔH> and <TΔS> are the mean values of enthalpic and entropic components of binding free energy, respectively; (ΔH)_STD_ and (TΔS)_STD_ are standard deviations corresponding to these values.

**Table 6 ijms-24-08083-t006:** Averages of the binding enthalpy for the amino acid residues of Mpro bound to the identified and control compounds.

Residue Contribution to the Binding Energy (kcal/mol) ^1,2,3^									
Residue of Mpro				Compounds					
	Inhibitor I	Inhibitor II	I	II	III	IV	V	VI	VII
Thr-25	-	−0.6 ± 0.4	-	-	−1.9 ± 0.9	−0.8 ± 0.4	-	−0.7 ± 0.6	−1.7 ± 0.4
Leu-27	−0.5 ± 0.3	−1.2 ± 0.4	-	-	−1.1 ± 0.3	−0.5 ± 0.2	−0.6 ± 0.2	−2.1 ± 0.7	−1.0 ± 0.2
**His-41**	**−2.2 ± 0.6**	**-**	**−2.1 ± 0.4**	**−0.9 ± 0.6**	**−0.7 ± 0.3**	**−0.9 ± 0.5**	**−1.4 ± 0.4**	**-**	**−1.2 ± 0.3**
Ser-46	-	-	-	-	-	−1.5 ± 0.9	−1.2 ± 0.4	-	−0.6 ± 0.4
**Met-49**	**−1.1 ± 0.6**	**−0.9 ± 0.6**	**−2.7 ± 1.5**	**−1.8 ± 0.9**	**−1.8 ± 0.5**	**−0.8 ± 0.3**	**−1.2 ± 0.3**	**−0.8 ± 0.6**	**−1.2 ± 0.3**
Leu-141	-	−0.5 ± 0.3	-	-	-	-	−1.1 ± 0.3	−0.6 ± 0.3	-
**Asn-142**	**−2.5 ± 0.6**	**−2.5 ± 0.6**	**-**	**-**	**−1.4 ± 0.7**	**−0.7 ± 0.8**	**−0.7 ± 0.5**	**−3.3 ± 1.2**	**-**
**Gly-143**	**−1.8 ± 0.3**	**−2.3 ± 0.5**	**-**	**-**	**−1.9 ± 0.4**	**-**	**−0.5 ± 0.2**	**−2.2 ± 0.6**	**-**
Ser-144	−0.7 ± 0.4	−1.0 ± 0.4	-	-	-	-	-	−1.7 ± 0.6	-
**Cys-145**	**−1.4 ± 0.3**	**−1.6 ± 0.5**	**-**	**-**	**−1.1 ± 0.3**	**−0.9 ± 0.3**	**−1.3 ± 0.3**	**−2.2 ± 0.7**	**−1.4 ± 0.3**
His-163	−1.7 ± 0.3	−1.7 ± 0.4	-	-	-	-	−1.2 ± 0.5	−0.6 ± 0.2	−1.5 ± 0.4
His-164	-	-	−0.5 ± 0.3	-	-	-	−2.9 ± 0.8	-	−0.9 ± 0.2
**Met-165**	**−2.6 ± 0.4**	**−2.6 ± 0.7**	**−2.9 ± 0.4**	**−2.5 ± 0.7**	**−3.0 ± 0.4**	**−2.9 ± 0.4**	**−3.5 ± 0.5**	**-**	**−3.7 ± 0.5**
**Glu-166**	**−1.4 ± 0.6**	**−1.2 ± 0.7**	**−1.0 ± 0.5**	**-**	**−2.6 ± 0.6**	**−1.6 ± 0.7**	**−2.0 ± 0.8**	**−0.6 ± 0.8**	**−1.1 ± 0.6**
Leu-167	-	-	−1.4 ± 0.5	−2.1 ± 0.6	−1.0 ± 0.4	−0.8 ± 0.2	−0.7 ± 0.4	-	-
Pro-168	-	-	−0.6 ± 0.3	−2.0 ± 0.5	−0.9 ± 0.4	−1.3 ± 0.3	−1.1 ± 0.4	-	-
Phe-185	-	-	-	−1.4 ± 0.5	-	-	-	-	-
**Asp-187**	−1.9 ± 0.7	−0.7 ± 0.9	−2.4 ± 0.4	−1.2 ± 0.7	-	-	-	-	−1.4 ± 0.3
Arg-188	-	-	−1.2 ± 0.6	−0.5 ± 0.5	-	-	-	-	−1.0 ± 0.4
**Gln-189**	**−1.0 ± 0.8**	**−1.3 ± 0.6**	**−2.5 ± 0.8**	**−2.9 ± 1.1**	**−2.9 ± 1.6**	**−3.3 ± 0.7**	**−0.7 ± 0.8**	**-**	**−1.1 ± 0.6**
Thr-190	-	-	−0.6 ± 0.4	−1.2 ± 0.8	-	−0.6 ± 0.2	-	-	-
Gln-192	-	-	-	−1.3 ± 0.6	-	−0.5 ± 0.2	-	-	-

^1^ Data for the Mpro residues with the binding energy ≤ −0.5 kcal/mol are presented. ^2^ The averages of the residue contributions to the binding energy and corresponding standard deviations are given. ^3^ The Mpro residues dominating the ligand/Mpro interaction are highlighted in bold.

**Table 7 ijms-24-08083-t007:** Values of RMSF for the Mpro residues contributing to the binding enthalpy.

Compounds									
Residue of Mpro	I	II	III	IV	V	VI	VII	Inhibitor I	Inhibitor II
	Values of RMSF (Å) for the Individual Residues of Mpro								
His41	0.5	0.5	0.6	0.6	0.6	0.6	0.5	0.7	0.6
Met49	2.0	1.0	1.3	1.1	1.1	2.0	0.9	2.2	2.0
Asn142	1.2	1.8	1.0	1.2	0.9	0.9	1.1	0.9	0.8
Gly143	1.0	0.9	0.8	0.7	0.8	0.9	0.9	0.7	0.7
Cys145	0.6	0.6	0.5	0.4	0.5	0.6	0.5	0.5	0.5
Met165	0.7	0.6	0.7	0.6	0.5	0.6	0.5	0.6	0.6
Glu166	0.8	0.7	0.8	0.6	0.6	0.8	0.6	0.7	0.7
Gln189	1.3	1.1	1.4	0.9	1.0	1.7	1.0	1.5	1.8

**Table 8 ijms-24-08083-t008:** Description of the investigated combinations of generative LSTM autoencoder models and generation modes.

Model	Generation Starting Point Description	Generation Process Description
Unsupervised (embeddings model)	Random number vectors drawn from fitted normal distributions	Random numbers are used as embeddings and fed to the decoder
Unsupervised (embeddings model)	Compounds with binding free energy less than −9 kcal/mol, sampled from the test set	Embeddings for these compounds are calculated, distortion is then added, and updated embeddings are fed to the decoder
Semi-supervised (energy model)	Random number vectors drawn from fitted normal distributions and a preset binding free energy value	Random vectors are used as embeddings and are passed as latent layer inputs along with a preset binding free energy value
Semi-supervised (energy model)	Compounds with binding free energy less than −8 kcal/mol were sampled from the test set and improved binding free energy values	Embeddings for these compounds are calculated, then distortion is added, and updated embeddings along with improved binding free energy values are passed to the decoder

## Data Availability

The generator part of the developed autoencoder for creating SMILES representations of chemical compounds with a preset value of binding free energy is available at https://github.com/Nikishul/smiles-GAE (accessed on 25 April 2023).

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
