# Peer review of "AI-Driven De Novo Design and Molecular Modeling for Discovery of Small-Molecule Compounds as Potential Drug Candidates Targeting SARS-CoV-2 Main Protease"

_ijms, 2023, doi:10.3390/ijms24098083_

Round 1

Reviewer 1 Report

The paper shows use of molecular modelling s for potential design of molecule compounds that can inhibit the catalytic activity of SARS-CoV-2 main protease (Mpro). Require further correction as shown below; 

60-68, require a reference addition

Fig 4, I to VII - The authors should redraw the structures in the more acceptable way with sugars shown in chair conformation

Fig 5, 1 to VII - The receptor-ligand interactions are not well visible. It would be better visible if you could attach a 2D diagram

202 207, State more explanation of these controls.

It would have been more effective if you used a known ligand as a control, that was reported to exhibit its activity via the same binding site

208-210 “within the 150 ns time domain”

In Figure 6 the experiment time is 200 ns, is the average calculated based on 150 ns or 200 ns?

231-235, “His-41 (compounds I−V, VII), Met-49 (compounds I-VII), Asn-142 (compounds III−VI), Gly-143 (compounds III−VI), Cys-145 (compounds III−VII), Met-165 (compounds I−V, VII), Glu-166 (compounds I, III−VII), Asp-187 (compounds I, II, VII), and Gln-189 (compounds I−V, VII) (Table 7).”

Met-49 is not included in Figure 5 VI. Show their position regarding to the binding pocket.

Met-49 is not included in Figure 5 VI. Show their position regarding to the binding pocket.

Cys-145 is not included in Figure 5 III, V and VII. Show their position regarding to the binding pocket

Asp-187 is not included in any of mentioned compounds in Figure 5. Show their position regarding to the binding pocket

245, Are these scores obtained from only one docking experiment ?

274 “orange line shows a simple moving average over the 20 ns time domain”

-          you mean 200 ns ?

277, Is there any explanation for the Structure-Activity Relationship?

334, “a set of 711 102”   & in other part of article etc

-          you mean 711,102 ?

414, “19 × 21 × 23 = 9177 Å3 “

Why this value for the parameters used etc   reference etc

Fig 1 2 & 3 & first table are numbered within last section 3 of this manuscript

Reviewer 2 Report

In the manuscript   “AI Driven De Novo Design and Molecular Modeling for Discovery of Small-Molecule Compounds as Potential Drug Candidates Targeting SARS-CoV-2 Main Protease”. Andrianov and colleagues identified new scaffolds to development drugs against SARS-CoV-2 Mpro through the new bioinformatics technologies.

Some points should be addressed by authors to improve the quality of manuscript.

1.       The introduction is very long. Please, revise this section and include the necessary background to get the objective of this work.

2.       It is not easy to see the importance of the collective citations selected when they are presented in the form (16-21). Individual expressions for each citation can improve the presentation of the work.

3.       My main criticism for this paper is that it does not include any biological validation.

4.       Among the main problems with chemical compounds are the potential cytotoxicity and its solubility in aqueous systems.  Could authors discuss the potential limitations of the new identified compounds?

Round 2

Reviewer 2 Report

Authors have revised my main concerns and the manuscript has been appropiately edited.